# The Circadian Rhythm of Intracellular Protoporphyrin IX Accumulation Through Heme Synthesis Pathway in Bladder Urothelial Cancer Cells Exposed to 5-Aminolevulinic Acid

**DOI:** 10.3390/cancers16234112

**Published:** 2024-12-08

**Authors:** Nobutaka Nishimura, Makito Miyake, Sayuri Onishi, Mitsuru Tomizawa, Takuto Shimizu, Kenta Onishi, Shunta Hori, Yosuke Morizawa, Daisuke Gotoh, Yasushi Nakai, Nobumichi Tanaka, Kiyohide Fujimoto

**Affiliations:** 1Department of Urology, Nara Medical University, 840 Shijo-cho, Kashihara 634-8522, Nara, Japan; nobunishimura@naramed-u.ac.jp.com (N.N.); makitomiyake@naramed-u.ac.jp (M.M.); sayuri3@naramed-u.ac.jp (S.O.); tomimit.com@naramed-u.ac.jp (M.T.); t-shimizu@naramed-u.ac.jp (T.S.); horimaus@naramed-u.ac.jp (S.H.); morizawa@naramed-u.ac.jp (Y.M.); dgotou@naramed-u.ac.jp (D.G.); nakaiyasushi@naramed-u.ac.jp (Y.N.); sendo@naramed-u.ac.jp (N.T.); 2Department of Prostate Brachytherapy, Nara Medical University, 840 Shijo-cho, Kashihara 634-8522, Nara, Japan

**Keywords:** circadian rhythm, protoporphyrin IX, 5-aminolevulinic acid, bladder cancer, photodynamic diagnosis, transurethral resection, clock genes, heme synthesis pathway

## Abstract

This study investigated the circadian rhythm of intracellular protoporphyrin IX (PPIX) accumulation in bladder cancer cells exposed to 5-aminolevulinic acid to understand the variability in red fluorescence intensity observed during photodynamic diagnosis-assisted transurethral resection among different tumors and patients. We identified the circadian rhythm of the clock genes *PER2* and *BMAL1* and their impact on PPIX accumulation in bladder cancer cell lines UM-UC-3 and J82, as well as in mouse xenograft models. Additionally, enzymes involved in heme synthesis, including coproporphyrinogen oxidase and ferrochelatase, were found to follow circadian rhythms. When red fluorescence intensity was quantified and compared among the four groups based on surgery start times, from 9 a.m. to 5 p.m., it peaked in surgeries performed between 3 and 5 p.m. These findings suggest that optimizing surgery timing based on circadian rhythms may improve the efficacy of photodynamic diagnosis in bladder cancer.

## 1. Introduction

Transurethral resection of bladder tumor (TURBT) has been the gold standard for the diagnosis and treatment of non-muscle invasive bladder cancer (NMIBC) for over 100 years [1,2,3]. TURBT is indicated as the initial treatment to ensure an accurate pathological diagnosis and complete resection of NMIBC. The procedural techniques for TURBT have significantly advanced over time, and photodynamic diagnosis-assisted TURBT (PDD-TURBT) was recently introduced for the treatment of NMIBC [4,5]. This modality enables surgeons to detect very small and/or flat lesions that may be overlooked by conventional white-light TURBT (WL-TURBT). The efficacy of PDD-TURBT is attributed to the intracellular accumulation of protoporphyrin IX (PPIX) caused by an overdose of oral 5-aminolevulinic acid hydrochloride (5-ALA), intravesical 5-ALA, or intravesical hexyl aminolevulinate (HAL) [6,7,8]. Many studies have demonstrated the oncologic efficacy and clinical safety of 5-ALA, which in Japan has been approved only for oral administration for PDD-TURBT [9,10,11].

5-ALA is metabolized by various enzymes in the mitochondria and cytoplasm of mammalian cells into heme as an intermediate, such as PPIX [12,13]. In cancer cells exposed to an overdose of 5-ALA, the downregulation of ferrochelatase (FECH) halts the conversion of PPIX to heme in the mitochondria, leading to an overaccumulation of intracellular PPIX [14,15], which is a photosensitizer that emits red fluorescence when exposed to light wavelengths of 430–435 nm [16,17]. Adenosine triphosphate-binding cassette sub-family G member 2 (ABCG2) is a transporter protein that modulates the intracellular levels of various substances, including PPIX, by actively transporting them out of the cell [18,19]. Overexpression of ABCG2 is associated with less PPIX accumulation, suggesting that drugs that decrease ABCG2 expression may enhance the efficacy of photodynamic therapy (PDT) [20]. Therefore, enzymes in the heme synthesis pathway are significantly associated with PPIX overaccumulation in cancer cells. One of the clinical limitations of PDD-TURBT is that some tumors are undetectable using red fluorescence due to the decreased accumulation of PPIX, even after exposure to 5-ALA [21]. However, the mechanism underlying the differences in fluorescence intensity among tumors remains unclear.

The circadian rhythm is an endogenous timekeeping mechanism that is present in almost all living organisms, from bacteria to humans [22]. The mammalian circadian rhythm involves both positive and negative regulators with a complex autoregulatory transcriptional and translational feedback loop. Savino et al. [23] demonstrated changes in the expression of clock genes associated with heme biosynthesis disorders. For example, the activation of ALA synthase (ALAS), an enzyme catalyzing the conversion of succinyl coenzyme A (CoA) and glycine to 5-ALA, is promoted by the *BMAL1/NPAS2* complex [24]. The transcription of *NPAS2* and *BMAL1* is also repressed by *REV-ERB*, the stability of which is influenced by heme binding [25].

We hypothesized, therefore, that the circadian rhythm may influence the intracellular accumulation of PPIX by regulating the heme synthesis pathway, resulting in varying tumor fluorescence intensities during PDD-TURBT. We further propose that the heme metabolism cycle, regulated by clock genes, may trigger intraday variations in fluorescence intensity as a result of the intracellular accumulation of PPIX, thereby impacting tumor detection rates.

## 2. Materials and Methods

### 2.1. Cell Lines and Synchronization by Forskolin (FSK)

We assessed the circadian rhythm of the intracellular accumulation of PPIX in the cultured urothelial cancer cells UM-UC-3 and J82, which were purchased from the American Type Culture Collection (Manassas, VA, USA). The clock genes in cultured cells are not synchronized because the culture conditions, such as temperature and CO_2_/O_2_ levels, remain constant, meaning factors that typically reset the circadian rhythm do not change throughout the day. Therefore, we synchronized the clock genes of these cells using FSK (Nacalai Tesque, Kyoto, Japan) [26], which activates the cyclic adenosine monophosphate (cAMP)/protein kinase A pathway, thereby increasing the response of cAMP element-binding protein activity at the PER1 promoter, which may, in turn, synchronize the circadian rhythm in urothelial cancer cells [27]. We periodically measured clock gene expression and intracellular concentrations of PPIX in FSK-activated urothelial cancer cells over a 24 h period.

The cell lines were maintained in Roswell Park Memorial Institute Medium 1640 (Nacalai Tesque, Kyoto, Japan) supplemented with 10% fetal bovine serum (FBS) (Nichirei Biosciences Inc., Tokyo, Japan), 100 U/mL penicillin, and 100 µg/mL streptomycin (Nacalai Tesque, Kyoto, Japan) in a standard humidified incubator at 37 °C with 5% CO_2_. UM-UC-3 and J82 cells were preincubated in six-well plates at a density of 0.3 × 10^6^ cells per well. The cells were activated with FSK (10 µM in 0.5% FBS) 24 h after seeding and exposed to 5-ALA 2 h after exposure, followed by the measurement of the intracellular concentration of PPIX 4 h later. The fluorescence intensity of PPIX was measured every 4 h for 24 h, and real-time reverse transcription polymerase chain reaction (RT-qPCR) was performed every 4 h to evaluate target gene expression.

### 2.2. Small Interfering Ribonucleic Acid (siRNA) Transfection

We prepared urothelial cancer cells with *PER2* and *BMAL1* knockdown to evaluate the changes in the intracellular concentration of PPIX. Three siRNAs targeting *PER2* (PER2-siRNA1, cat. ID. s16930; PER2-siRNA2, cat. ID. s16931; and PER2-siRNA3, cat. ID. s16932) and three siRNAs targeting *BMAL1* (BMAL1-siRNA1, cat. ID. s1616; BMAL1-siRNA2, cat. ID. s1617; and BMAL1-siRNA3, cat. ID. s1618) as well as a negative control siRNA (NC-siRNA) with a scrambled sequence that did not match any human mRNAs were synthesized by Life Technologies (Carlsbad, CA, USA). Using multiple siRNAs for each target gene helped minimize the potential impact of off-target effects. Cells were seeded in 6-well plates at a density of 0.3 × 10^6^ cells per well, and siRNAs targeting *PER2* and *BMAL1* were prepared at a concentration of 100 nM. The Lipofectamine RNAiMAX Reagent (Applied Biosystems, Foster City, CA, USA) was used according to the manufacturer’s protocol. The cells were transfected using the reverse transfection technique, in which they were simultaneously transfected and plated. To mitigate the cytotoxic effects of the transfection reagent, the medium was replaced with a fresh growth medium 24 h after the siRNA Transfection.

### 2.3. Animal Experiments

All animal experiments were approved by the Committee on Animal Research at Nara Medical University (approval no.: 12670) and conducted in accordance with the Guidelines for the Welfare of Animals for Experimental Neoplasia.

We purchased twenty-one 4- to 6-week-old male BALB/c-nu/nu mice from CLEA Japan (Tokyo, Japan) that were housed (four per cage) under a 12 h light/dark cycle (lights on from 8 a.m. to 8 p.m. and off from 8 p.m. to 8 a.m.) with free access to food and water. The sample size was set at 3 mice per time point, totaling 21 mice divided among seven groups (Zeitgeber time [ZT] 0, 4, 8, 12, 16, 20, 24). Each mouse was subcutaneously inoculated in the inguinal region with 2.0 × 10^6^ UM-UC-3 cells using Matrigel (Life Technologies, Carlsbad, CA, USA). The first group received oral 5-ALA (400 mg/kg) 14 days post-inoculation, and 4 h later (ZT 0), the tumors were removed from these three mice under anesthesia. PPIX was extracted from the tumors, and its intracellular concentration was quantified, as well as the mitochondrial RNA (mRNA) levels of *PER2* and *BMAL1*. The next group (ZT 4) received the same initial treatment, with the procedure repeated 4 h later. For each subsequent group, the initial procedure was the same, while the procedure was repeated one additional time over the previous group. Finally, the tumors were removed at 7 time points every 4 h (ZT 0, 4, 8, 12, 16, 20, and 24), with the intracellular concentrations of PPIX and mRNA levels of *PER2* and *BMAL1* measured at each time point. In this experiment, the start time (ZT 0) was 8 a.m., which indicated the beginning of the light phase of the diurnal cycle.

Eight enzymes that convert 5-ALA to heme or PPIX are exported outside the cells, including 5-ALA dehydratase (ALAD), porphobilinogen deaminase (HMBS), uroporphyrinogen III synthase (UROS), uroporphyrinogen decarboxylase (UROD), coproporphyrinogen oxidase (CPOX), protoporphyrinogen oxidase (PPOX), FECH, and ABCG2; therefore, we focused on these to identify the stages or enzymes most closely associated with clock genes. All eight enzymes were evaluated, along with intracellular PPIX accumulation, to determine whether they followed a circadian rhythm. RT-qPCR was performed using the complementary deoxyribonucleic acid (cDNA) obtained from previous in vivo studies.

### 2.4. RNA Extraction and RT-qPCR

Total RNA was extracted from the cell and tumor samples and homogenized using an RNeasy Mini Kit (Qiagen, Hilden, Germany). cDNA was synthesized using a High-Capacity cDNA Reverse Transcription Kit (Applied Biosystems, Foster City, CA, USA), and RT-qPCR was performed using a CFX Duet Real-time PCR system (Bio-Rad Laboratories, Hercules, CA, USA). TaqMan probes with the following IDs were obtained from Thermo Fisher Scientific (Waltham, MA, USA) and used for the quantification of target gene expression: Hs01007553_m1 (for *PER2*), Hs00154147_m1 (for *BMAL1*), Hs00765604_m1 (for *ALAD*), Hs00609296_g1 (for *HMBS*), Hs03405152_m1 (for *UROS*), Hs01099757_g1 (for *UROD*), Hs01071019_m1 (for *CPOX*), Hs00970229_g1 (for *PPOX*), Hs01555261_m1 (for *FECH*), and Hs01053790_m1 (for *ABCG2*). The mRNA levels of the target genes were quantified and calculated using gene-specific primers, and *GAPDH* (ID: Hs02786624_g1) was used as the housekeeping gene. Data analysis was performed using CFX Maestro software (version 2.0; Bio-Rad Laboratories, Hercules, CA, USA).

### 2.5. Measurement of Intracellular Concentration of PPIX

PPIX was extracted using a methanol-water system [28]. Four hours post-5-ALA exposure, the cells were washed twice with phosphate-buffered saline and trypsinized. The cell pellet was treated with 90% methanol (*v*/*v*) and sonicated for 20 min. Following centrifugation at 12,000× *g* for 15 min, the fluorescence intensity of the supernatant was measured using a microplate reader (TECAN, Männedorf, Switzerland). PPIX in 90% methanol was excited at 435 nm and emitted a fluorescence, which peaked at 630 nm. Commercial PPIX (BIOMOL, Philadelphia, PA, USA) dissolved in 90% methanol was used as the standard for calibrating the fluorescence intensity. For normalization, the intracellular concentration of PPIX in each sample was divided by the cell number before extraction, and the results were expressed as nanograms of PPIX per cell (ng/cell). Additionally, the intracellular concentration of PPIX in mouse tumors was calculated as nanograms per milligram of tumor tissue (ng/mg tissue).

### 2.6. Fluorescence Intensity Images from PDD-TURBT in Human Experiments

All procedures involving human participants were performed in accordance with the ethical standards of the Nara Medical University Ethical Committee (approval no.: 1256) and the 1964 Declaration of Helsinki and its later amendments or comparable ethical standards.

We obtained images during each PDD-TURBT procedure and assessed the differences in the fluorescence intensity among these images. A total of 70 patients who underwent PDD-TURBT for NMIBC 2–4 h post-5-ALA administration between July 2023 and January 2024 at Nara Medical University were initially included in this study. Images in which the background brightness of the white bladder mucosa could be standardized were selected for analysis. The brightness level was quantified using ImageJ freeware (National Institutes of Health, Rockville, MD, USA) with JavaTM version 1.8.0_112 (64-bit) on Windows 11 [29]. Of the 70 patients, 11 were excluded due to discrepancies in the brightness of the background bladder mucosa compared to that of other patients. The red fluorescence intensities on the images captured during PDD-TURBT were quantified and compared among the four groups according to the time of surgery: 9–11 a.m., 11 a.m.–1 p.m., 1–3 p.m., and 3–5 p.m.

### 2.7. Statistical Analysis

Graphs were constructed using GraphPad Prism 7.0 software (GraphPad Software, San Diego, CA, USA), and statistical analyses of patient characteristics were conducted using appropriate tests, including the Student’s *t*-test and the chi-squared test, with EZR software ver. 1.65 (Saitama Medical Center, Jichi Medical University, Saitama, Japan) [30]. Circadian rhythm data were fitted using a sinusoidal model, and the goodness of fit was evaluated using the coefficient of determination (R^2^). To categorize the strength of the fit, the following criteria were used: strong fit, R^2^ ≥ 0.80; moderate fit, R^2^ = 0.50–0.79; and weak fit, R^2^ value < 0.50. Statistical significance was set at *p* < 0.05.

## 3. Results

### 3.1. Synchronization of Expression of Clock Genes in Bladder Cancer Cells

Appendix A shows the time-course change in the mRNA levels of *PER2* and *BMAL1* over a 24 h period in FSK-treated- and non-treated cells. A circadian clock gene expression curve was observed in FSK-treated UM-UC-3 and J82 cells. Although non-FSK-treated cells showed a circadian curve for clock gene expression, a higher amplitude and clearer periodic curve were observed in FSK-treated cells. The R^2^ value of the fitted curve for FSK-treated cells was higher than that for non-FSK-treated cells, indicating a strong or moderate fit to the sinusoidal model. Furthermore, the circadian curves of *PER2* and *BMAL1* seemed to exhibit the opposite trends.

### 3.2. Circadian Rhythm of Intracellular Accumulation of PPIX In Vitro

Figure 1A shows the experimental scheme for monitoring the time-course change in the intracellular PPIX concentration and the mRNA levels of *PER2* and *BMAL1* over the course of 24 h. The circadian curve for the intracellular PPIX concentration was observed in both UM-UC-3 and J82 cells (Figure 1B,D), along with rhythmic gene expression of *PER2* and *BMAL1* (Figure 1C,E). The R^2^ values of the fitted curves for the intracellular accumulation of PPIX in UM-UC-3 and J82 cells showed a weak or moderate fit to the sinusoidal model (R^2^ value, 0.35 and 0.55, respectively).

### 3.3. Circadian Rhythm of the Intracellular Accumulation of PPIX and Clock Genes

Figure 2 shows the time-course change in the intracellular concentration of PPIX in bladder cancer cells in which *PER2* and *BMAL1* were knocked down using siRNA transfection. However, the R^2^ value for the fitted curve for the intracellular accumulation of PPIX in NC-siRNA-transfected UM-UC-3 cells showed a strong or moderate fit to the sinusoidal model (R^2^ = 0.97 and 0.70, respectively). On the other hand, no circadian rhythm was observed in PER2- or BMAL1-siRNA-transfected UM-UC-3 and J82 cells. The relative mRNA levels of *PER2* and *BMAL1* were consistently suppressed throughout 24 h by all siRNAs in both UM-UC-3 and J82 cells.

### 3.4. Circadian Rhythm of Intracellular PPIX Accumulation in Mouse Xenograft Tumors

Figure 3A shows the experimental schema used to monitor the time-course change in the intracellular concentration of PPIX and the mRNA levels of *PER2* and *BMAL1* in mouse xenograft tumors over 24 h. Figure 3B shows that the R^2^ value of the fitted curve for the intracellular accumulation of PPIX in the xenograft indicated a moderate fit to the sinusoidal model (R^2^ = 0.74). Figure 3C shows that the R^2^ value for the fitted curve of mRNA levels of *PER2* and *BMAL1* in xenografts indicated a moderate or strong fit to the sinusoidal model (R^2^ = 0.66 and 0.81, respectively). Furthermore, the intracellular PPIX concentration and mRNA levels of *BMAL1* reached their highest levels during the dark phase of the diurnal cycle.

### 3.5. Enzymes in the Heme Synthesis Pathway Associated with the Circadian Rhythm in Intracellular Accumulation of PPIX

Figure 4A shows the heme synthesis pathway, including several enzymes and metabolites from succinyl-CoA, glycine, and 5-ALA to heme. Figure 4B–I show the time-course change of intracellular PPIX concentration and mRNA expression of eight enzymes in xenograft tumors. The mRNA expression levels of *CPOX* and *FECH* had circadian curves that showed the opposite of those for intracellular PPIX concentration. The R^2^ value for the fitted curve in mRNA levels of *CPOX* and *FECH* in xenograft, respectively, indicated a moderate fit to the sinusoidal model (R^2^ = 0.73 and 0.75, respectively).

### 3.6. Difference in Red Fluorescence Intensity of Images Obtained During PDD-TURBT

Table 1 shows the clinicopathological variables of patients who underwent PDD-TURBT and their categorization according to the start time of surgery. The four groups consisted of 17 patients who started PDD-TURBT between 9 and 11 a.m., 17 who started between 11 a.m. and 1 p.m., 14 who started between 1 and 3 p.m., and 11 who started PDD-TURBT between 3 and 5 p.m. No significant differences were noted in any of the clinicopathological variables among the four groups. Figure 5 shows the flowchart of patient selection, the actual reconstructed images using the ImageJ software ver. 1.54i, and the comparison of the fluorescence intensity of images among the four groups. The fluorescence intensity showed a significant diurnal difference, with the highest level observed in patients treated between 3 and 5 p.m.

## 4. Discussion

Hemes are essential molecules involved in various biological processes, including oxygen transport, electron transfer, and catalysis. The heme synthesis pathway consists of several steps, which start in the mitochondria with the condensation of glycine and succinyl-CoA to form δ-ALA, catalyzed by *ALAS-1* [31]. ALA is then transported to the cytoplasm, where it undergoes several enzymatic transformations to form PPIX, which eventually returns to the mitochondria, where FECH adds iron to form heme. Circadian genes influence the expression of enzymes involved in heme synthesis and degradation. For example, expression of *ALAS-1*, a key enzyme in heme synthesis, is regulated by the circadian rhythm. Heme itself can regulate circadian gene expression by binding to nuclear receptors such as *REV-ERBα*, which in turn represses the transcription of *BMAL1*, influencing the circadian rhythm [24,32].

We evaluated the association between the intermediate metabolite PPIX and clock genes in urothelial cancer cells. The circadian rhythm of intracellular PPIX accumulation was lost when UM-UC-3 and J82 cells were transfected with PER2- and BMAL1-siRNAs (Figure 2). In both UM-UC-3 and J82 cells, the circadian rhythm of intracellular PPIX accumulation was not observed in either siRNA group. These results suggest that *PER2* and *BMAL1* might be strongly associated with the circadian rhythm of intracellular PPIX accumulation. Previous studies demonstrated that the *BMAL1/NPAS2* complex directly regulates the expression of *ALAS-1* [24]. Activation of *BMAL1*, therefore, may regulate the intraday shift in the intracellular accumulation of PPIX.

Differences in the intracellular concentration of PPIX can induce changes in the fluorescence intensity during PDD-TURBT, potentially affecting its performance [33]. For example, in Figure 3, during the dark phase, intracellular PIPX concentration gradually increased, reaching its highest level when *BMAL1* was also at its peak. This pattern was consistent with the results of previous studies. In nocturnal animals such as mice, *BMAL1* expression is highest at night, which corresponds to their active phase. Conversely, in diurnal animals such as humans, *BMAL1* expression peaks during the day, aligning with the period of activity [34]. These results suggest that PPIX had the highest concentration during the active phase. Furthermore, we believe that the novel findings can be extended to other modalities of PDD using 5-ALA, not only for the detection of bladder cancer but also for other malignancies, such as prostate and penile cancers, as well as pediatric surgeries [35]. While further evaluation of the circadian rhythm of PPIX in other cancers is warranted, we anticipate that the observed time-dependent variations in fluorescence intensity could also be applied in these contexts.

In the in vivo experiment, the intracellular PPIX concentration was higher at the end of the dark phase than at the beginning (Figure 3). Additionally, the fluorescence intensity of PPIX in the four groups assigned by two-hour intervals was higher at the end of the day than at the beginning (Figure 5). These results suggest that performing PDD-TURBT in the evening may be more effective than in the early morning. However, the difference in the fluorescence intensity of PPIX did not directly correlate with the brightness. Therefore, the differences in red light emission from tumors throughout the day during PDD-TURBT are unclear. The method for measuring fluorescence intensity is not concrete or reproducible, and we should evaluate the detection rate for bladder cancers at different times of the day. Therefore, in the future, we plan to conduct clinical trials to assess intraday differences in bladder cancer detection rates. If our hypothesis that the evening is preferable to the early morning for PDD-TURBT is proven, the timing of PDD-TURBT procedures may need to be reconsidered.

*FECH* and *CPOX* were associated with rhythmic patterns similar to the circadian curve of the intracellular PPIX concentration. Interestingly, the circadian curve of *FECH* mRNA levels was opposite to that of the intracellular concentration of PPIX. The circadian rhythm of intracellular PPIX accumulation, therefore, may be influenced by the circadian rhythm of *FECH*, which is consistent with previous reports. Regulation of *FECH* expression affects PPIX levels in various diseases, including protoporphyria and bladder cancer [36,37]. When the mRNA levels of *FECH* were low, the intracellular concentration of PPIX was high and vice versa. These results may help investigate the detailed mechanisms in the future. However, the association between clock genes and the enzymes involved in the heme synthesis pathway, such as *FECH* and *CPOX*, remains unclear. Uncovering the relationship between these enzymes and clock genes may lead to more efficient PDD and PDT methods.

This study had several limitations. First, these findings were observed in only two urothelial cancer cell lines, which were selected based on their high tumorigenicity and ease of engraftment in nude mice. As a result, the reproducibility of these results may be limited. To address this, future studies should include a broader range of cell lines and involve a prospective design to evaluate differences in intratumoral PPIX concentrations and sensitivities across various models. Second, the sample size for each measurement of the intracellular PPIX concentration and mRNA levels was only three. Although we can calculate the mean and standard deviation [SD] and perform statistical analyses with the samples, the sample size is the minimum necessary, which affects data reliability. However, increasing the sample size is challenging and complicated because of the technical procedures involved. Third, the method used to measure the intracellular PPIX concentration had its own limitations. The deviation in the intracellular PPIX concentration was large (Figure 1), which may have reduced data reliability. Fourth, we evaluated the mRNA levels of *PER2* and *BMAL1* every 4 h. A bioluminescence recording method is available for the constant evaluation of gene expression [38,39]. For example, introducing luciferase into target cells or tissues and quantitatively recording bioluminescence based on its activity over time is a reproducible method for evaluating clock gene expression; however, our institution does not have the necessary equipment to record bioluminescence. Fifth, the number of patients who underwent PDD TURBT was limited, as samples were only available from July 2023, owing to storage regulations at our institution. Therefore, this may have resulted in a selection bias.

## 5. Conclusions

We identified the circadian rhythms of intracellular PPIX accumulation and the influence of clock genes *PER2* and *BMAL1*, which may contribute to the variability in red fluorescence intensity observed during PDD-TURBT. Our findings suggest that optimizing the timing of the photodynamic diagnosis in PDD-TURBT according to the circadian rhythms could improve tumor detection and treatment outcomes.

## Figures and Tables

**Figure 1 cancers-16-04112-f001:**
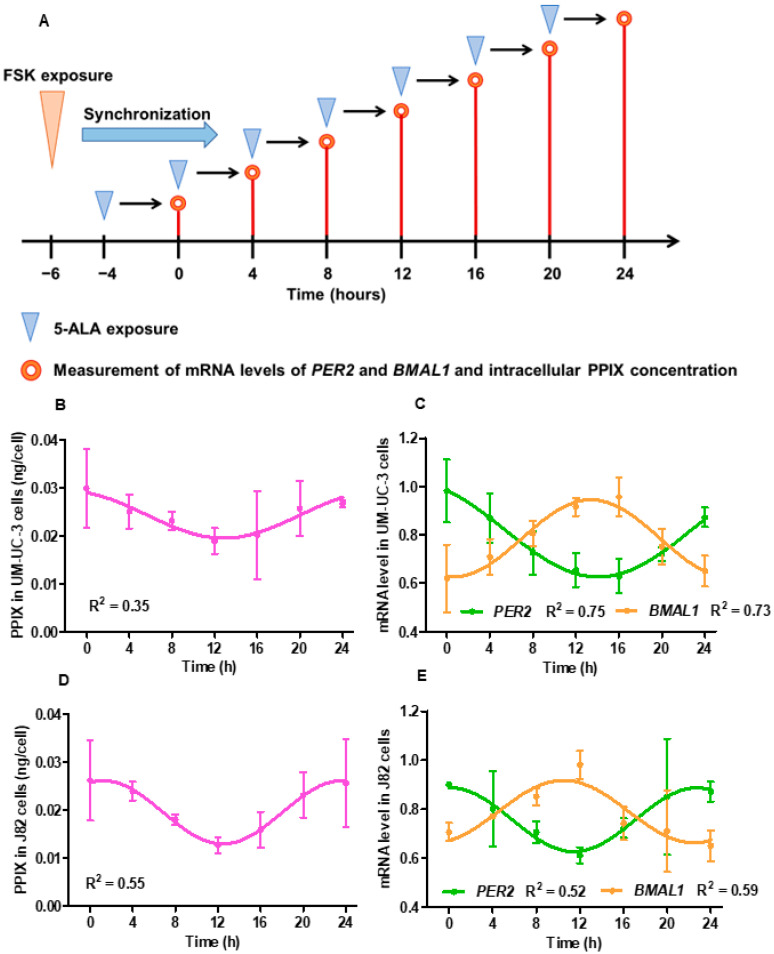
Circadian rhythm of intracellular accumulation of protoporphyrin IX (PPIX) in vitro. (**A**) shows the methods for measuring the intracellular concentration of PPIX and mitochondrial ribonucleic acid (mRNA) levels of *PER2* and *BMAL1.* Two bladder urothelial cancer cell lines, UM-UC-3 and J82, were exposed to forskolin (FSK) (10 µM in 0.5% fetal bovine serum [FBS]) 24 h after seeding, with 5-ALA exposure initiated 2 h post-FSK exposure. Four hours post-5-ALA exposure, the intracellular PPIX concentration was measured. Similarly, mRNA levels of *PER2* and *BMAL1* and intracellular PPIX concentration were measured every 4 h over a 24 h period, as shown in the schema. (**B**,**C**) show the shifts in the intracellular concentration of PPIX and mRNA levels, respectively, of *PER2* and *BMAL1* in UM-UC-3 cells. (**D**,**E**) show the shifts of intracellular concentration of PPIX and mRNA levels, respectively, of *PER2* and *BMAL1* in J82 cells.

**Figure 2 cancers-16-04112-f002:**
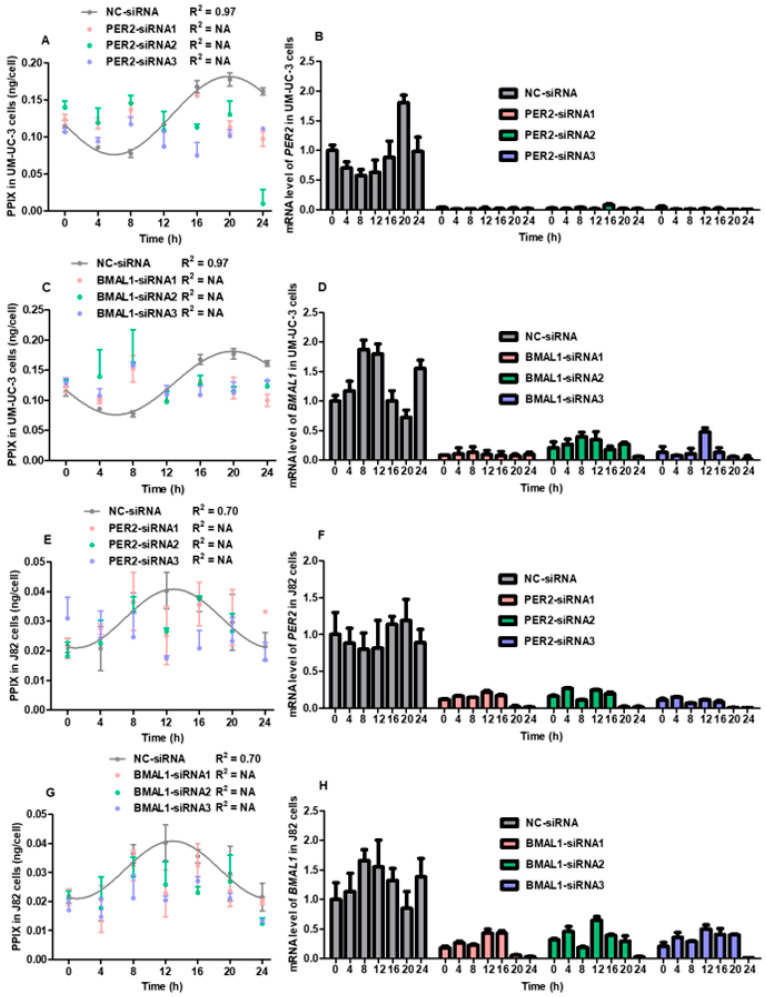
Association between the circadian rhythm of intracellular accumulation of PPIX and clock genes *PER2* and *BMAL1.* UM-UC-3 and J82 cells were exposed to FSK, and two hours later, the cells were transfected with PER2 and BMAL1-small interfering ribonucleic acids (siRNAs). After Transfection, the intracellular PPIX concentration was measured every four h for up to 24 h. (**A**) shows the shifts in the intracellular concentration of PPIX in UM-UC-3 cells transfected with three PER2-siRNAs and NC-siRNA. (**B**) shows the relative mRNA levels of *PER2* in UM-UC-3 cells transfected with three PER2-siRNAs and NC-siRNA. (**C**) shows the shifts in the intracellular concentration of PPIX in UM-UC-3 cells transfected with three BMAL1-siRNAs and NC-siRNA. (**D**) shows the relative mRNA levels of *BMAL1* in UM-UC-3 cells transfected with three BMAL1-siRNAs and NC-siRNA. (**E**) shows the shifts in the intracellular concentration of PPIX in J82 cells transfected with three PER2-siRNAs and NC-siRNA. (**F**) shows the relative mRNA levels of *PER2* in J82 cells transfected with three PER2-siRNAs and NC-siRNA. (**G**) shows the shifts in the intracellular concentration of PPIX in J82 cells transfected with three BMAL1-siRNAs and NC-siRNA. (**H**) shows the relative mRNA levels of *BMAL1* in J82 cells transfected with three BMAL1-siRNAs and NC-siRNA. Relative mRNA levels were normalized to the 0-h time point of the NC-siRNA group, which served as the blank control for calculation. The goodness-of-fit values that could not be calculated were represented as “NA”, indicating “not applicable”.

**Figure 3 cancers-16-04112-f003:**
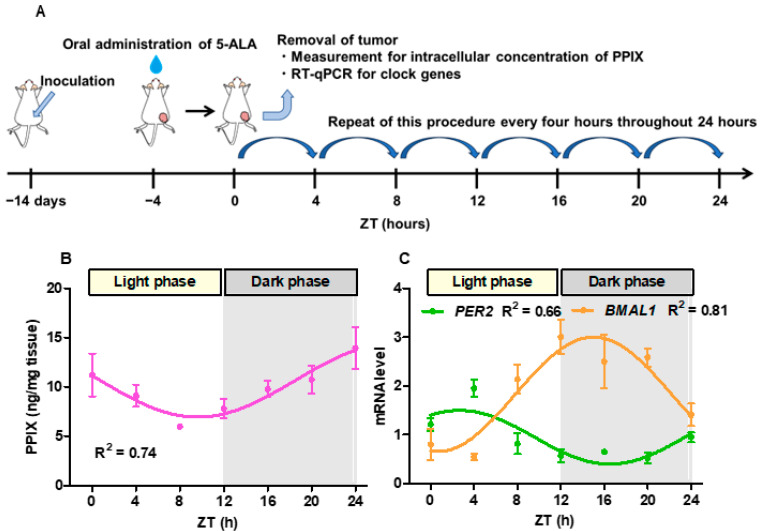
Circadian rhythm of intracellular accumulation of PPIX within mouse tumors. (**A**) shows the methods of measuring the intracellular PPIX concentration and mRNA levels of *PER2* and *BMAL1* in mouse tumors. UM-UC-3 cells were subcutaneously inoculated into twenty-one 4- to 6-week-old male BALB/c-nu/nu mice. The intracellular PPIX concentration and mRNA levels of *PER2* and *BMAL1* in the tumors grown in these mice were measured every 4 h for up to 24 h, as shown in the schema. The light and dark phases were switched every 12 h (lights were switched on from 8 a.m. to 8 p.m. and off from 8 p.m. to 8 a.m.). The experiment started at Zeitgeber Time (ZT) 0, which marks the start of the light phase of the diurnal cycle. (**B**,**C**) show the shifts in the intracellular PPIX concentration and mRNA levels of *PER2* and *BMAL1*.

**Figure 4 cancers-16-04112-f004:**
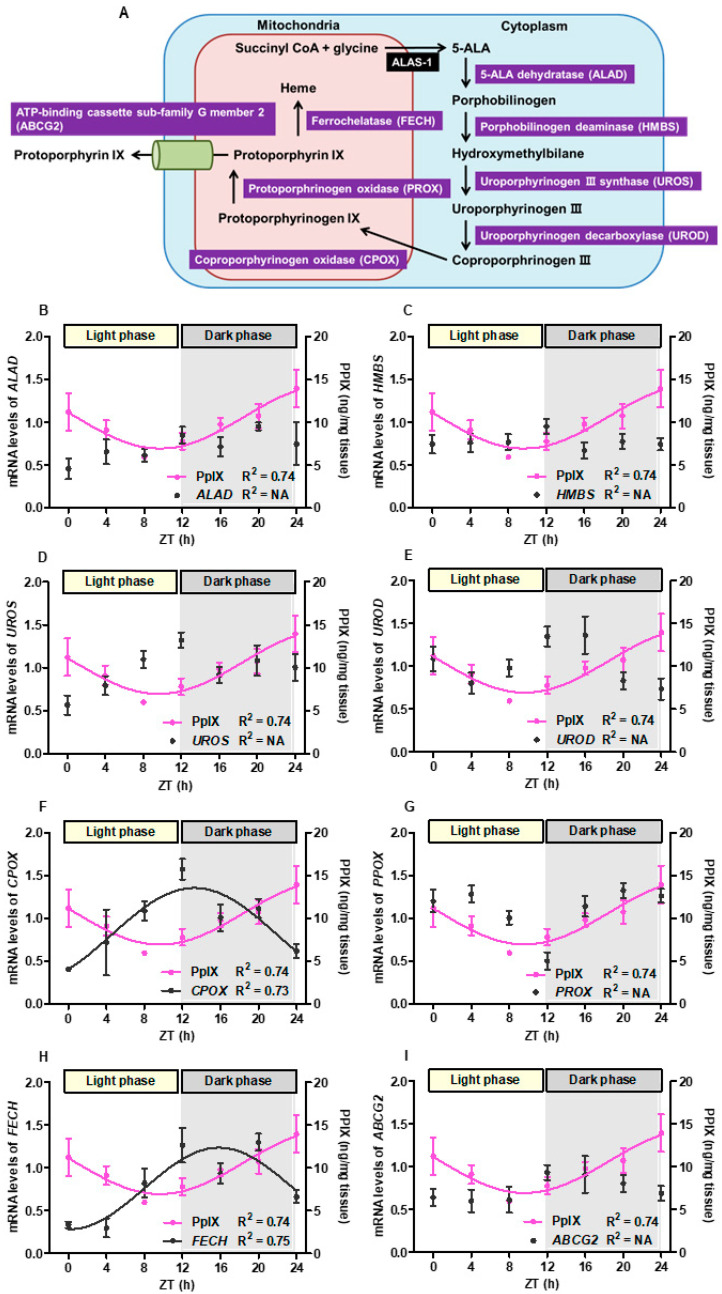
Association between intracellular PPIX accumulation and eight enzymes in the heme synthesis pathway. Eight main enzymes are involved in the heme synthesis pathway from 5-ALA to heme or PPIX export from cells (**A**). Tumors obtained from nude mice were used to evaluate the intracellular concentration of PPIX and the expression of genes related to heme synthesis. When mRNA levels exhibited a circadian rhythm, a sinusoidal curve was observed. (**B**–**I**) show the shifts in the intracellular PPIX concentration and mRNA levels for each target gene: *ALAD* (**B**), *HMBS* (**C**), *UROS* (**D**), *UROD* (**E**), *CPOX* (**F**), *PPOX* (**G**), *FECH* (**H**), and *ABCG2* (**I**). The goodness-of-fit values that could not be calculated were represented as “NA”, indicating “not applicable”.

**Figure 5 cancers-16-04112-f005:**
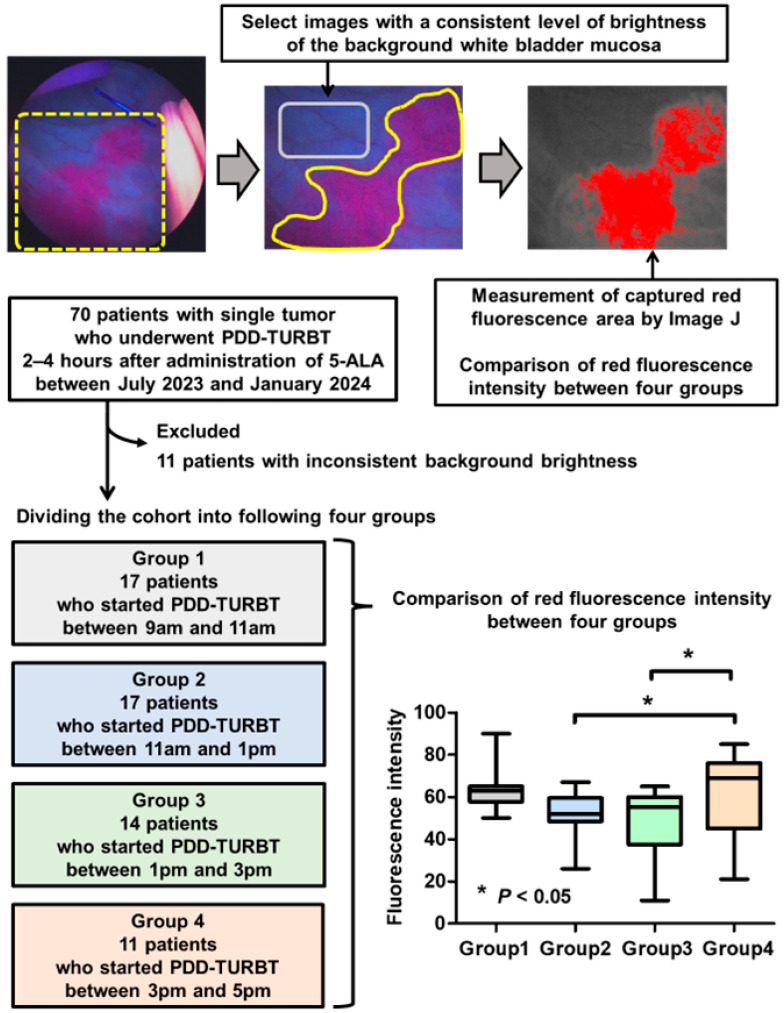
Change in fluorescence intensity of images obtained during photodynamic diagnosis-assisted transurethral resection of bladder tumor (PDD-TURBT) over time. The images captured PDD-TURBT were reconstructed using ImageJ software ver. 1.54i (National Institutes of Health, Rockville, MD, USA), and a flowchart of patient selection is shown [29]. Images where the background brightness of the white bladder mucosa could be standardized were selected, and 11 patients were excluded due to discrepancies in the brightness of the background bladder mucosa compared to that of the others. Among the 70 patients with single tumors who underwent PDD-TURBT 2–4 h after the administration of 5-ALA between July 2023 and January 2024, the following four groups were identified for the comparison of red fluorescence intensity: 17 patients who started PDD-TURBT between 9 and 11 a.m. (Group 1), 17 who started between 11 a.m. and 1 p.m. (Group 2), 14 who started between 1 and 3 p.m. (Group 3), and 11 who started between 3 and 5 p.m. (Group 4).

**Table 1 cancers-16-04112-t001:** Clinicopathological variables of patients who underwent PDD-TURBT in the morning or afternoon.

Variables	All	Group 1	Group 2	Group 3	Group 4	*p*-Value
n	59	17	17	14	11	
Start time of PDD-TURBT	–	9 a.m. to 11 a.m.	11 a.m. to 1 p.m.	1 p.m. to 3 p.m.	3 p.m. to 5 p.m.	–
Age, years, mean ± SD	75.6 ± 7.4	75.7 ± 6.7	75.9 ± 9.6	76.6 ± 5.5	73.6 ± 7.3	0.8 ^†^
Sex (%)						0.54 ^‡^
Male	51 (86.4)	15 (88.2)	13 (76.5)	13 (92.9)	10 (90.9)	
Female	8 (13.6)	2 (11.8)	4 (23.5)	1 (7.1)	1 (9.1)	
Size (%)						0.33 ^‡^
<30 mm	47 (79.7)	14 (82.4)	13 (76.5)	13 (92.9)	7 (63.6)	
≥30 mm	12 (20.3)	3 (17.6)	4 (23.5)	1 (7.1)	4 (36.4)	
T (%)						0.17 ^‡^
Ta	44 (74.6)	11 (64.7)	16 (94.1)	10 (71.4)	7 (63.6)	
T1	15 (25.4)	6 (35.3)	1 (5.9)	4 (28.6)	4 (36.4)	
Grade1973 (%)						0.61 ^‡^
G1	3 (5.1)	1 (5.9)	1 (5.9)	1 (7.1)	0 (0.0)	
G2	41 (69.5)	9 (52.9)	13 (76.5)	10 (71.4)	9 (81.8)	
G3	11 (18.6)	6 (35.3)	1 (5.9)	2 (14.3)	2 (18.2)	
Unknown	4 (6.8)	1 (5.9)	2 (11.8)	1 (7.1)	0 (0.0)	
Grade2004 (%)						0.11 ^‡^
Low grade	43 (72.9)	10 (58.8)	16 (94.1)	10 (71.4)	7 (63.6)	
High grade	16 (27.1)	7 (41.2)	1 (5.9)	4 (28.6)	4 (36.4)	
LVI (%)						0.48 ^‡^
No	57 (96.6)	16 (94.1)	17 (100.0)	14 (100.0)	10 (90.9)	
Yes	2 (3.4)	1 (5.9)	0 (0.0)	0 (0.0)	1 (9.1)	

LVI—lymphovascular invasion; PDD-TURBT—photodynamic diagnosis-assisted transurethral resection of bladder cancer; SD—standard deviation. †—Student’s *t*-test. ‡—Chi-square test.

## Data Availability

The data presented in this study are available on request from the corresponding author.

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
