# Peer review of "The Circadian Rhythm of Intracellular Protoporphyrin IX Accumulation Through Heme Synthesis Pathway in Bladder Urothelial Cancer Cells Exposed to 5-Aminolevulinic Acid"

_cancers, 2024, doi:10.3390/cancers16234112_

Round 1
Reviewer 1 Report
Comments and Suggestions for Authors
The topic is of interest and the research idea is new and original. Potentially, it has several implications with clinical practice in order to optimize the timing of diagnosis for Bca.
The aim of the study is clear. The methods well described. No ethical concerns. The study has been approved by the local committee. Limitations of the study are well declared.
I would only insert a brief perspective point of view on possible application of this findings in other settings (i.e. other urological tumors and tracers - i.e. Paraboschi I, Farneti F, Jannello L, et al. Narrative review on applications of fluorescence-guided surgery in adult and pediatric urology. AME Med J 2022;7:15).
Author Response
Comment: The topic is of interest and the research idea is new and original. Potentially, it has several implications with clinical practice in order to optimize the timing of diagnosis for Bca.
The aim of the study is clear. The methods well described. No ethical concerns. The study has been approved by the local committee. Limitations of the study are well declared.
I would only insert a brief perspective point of view on possible application of this findings in other settings (i.e. other urological tumors and tracers - i.e. Paraboschi I, Farneti F, Jannello L, et al. Narrative review on applications of fluorescence-guided surgery in adult and pediatric urology. AME Med J 2022;7:15).
Reply:
Thank you very much for your thoughtful feedback. I sincerely appreciate your consideration in accepting our study for this journal. As you pointed out, we have incorporated the recommended review into our manuscript and have revised the discussion section accordingly (page 24, lines 367–371).
Reviewer 2 Report
Comments and Suggestions for Authors
The manuscript titled "The Circadian Rhythm of Intracellular Protoporphyrin IX Accumulation Through Heme Synthesis Pathway in Bladder Urothelial Cancer Cells Exposed to 5-Aminolevulinic Acid" is highly interesting and well-written. It draws attention to the potential improvement of a technique already widely used in daily clinical practice. The application of 5-ALA for bladder cancer diagnosis is well-established; however, the authors propose an innovative approach by examining the circadian rhythm of cancer cells as a possible factor to optimize the timing of 5-ALA administration, thereby enhancing cancer detection.
The tables and figures are of excellent quality, effectively supporting the findings and facilitating reader comprehension. Given the manuscript’s relevance, clarity, and potential impact, I recommend accepting the paper in its current form.
Author Response
Comment: The manuscript titled "The Circadian Rhythm of Intracellular Protoporphyrin IX Accumulation Through Heme Synthesis Pathway in Bladder Urothelial Cancer Cells Exposed to 5-Aminolevulinic Acid" is highly interesting and well-written. It draws attention to the potential improvement of a technique already widely used in daily clinical practice. The application of 5-ALA for bladder cancer diagnosis is well-established; however, the authors propose an innovative approach by examining the circadian rhythm of cancer cells as a possible factor to optimize the timing of 5-ALA administration, thereby enhancing cancer detection.
The tables and figures are of excellent quality, effectively supporting the findings and facilitating reader comprehension. Given the manuscript’s relevance, clarity, and potential impact, I recommend accepting the paper in its current form.
Reply:
Thank you very much for your valuable comment. I sincerely appreciate your suggestion regarding the acceptance of our manuscript for this journal. In response to the feedback from other reviewers, we have made further modifications to the manuscript. We kindly ask you to review the revised version at your convenience.
Reviewer 3 Report
Comments and Suggestions for Authors
The present study has revealed that the circadian rhythm affects intracellular protoporphyrin IX (PPIX) accumulation in bladder cancer cells to 5-aminolevulinic acid. Moreover, the authors show that the fluorescence intensity was changed depending on timing in patients with bladder cancer who undertook photodynamic diagnosis-assisted transurethral resection of bladder tumor (PDD-TURBT). It would bring significant insights to clinical practice for bladder cancer. However, there are some concerns to be addressed.
1. Why did you select UM-UC-3 and J82 cells? These cell lines were derived from patients with muscle-invasive bladder cancer (NMIBC). PDD-TURBT is especially useful for detecting non-muscle invasive bladder cancer (MIBC).
2. Figure 4 suggests that the intracellular PPIX accumulation is associated with CPOX and FECH levels through circadian rhythm. Did the authors evaluate the association between PER2/BMAL1 and CPOX/FECH levels in vitro?
3. The previous studies have shown that the circadian rhythm is impaired in cancer. Why were the fluorescence intensities highest in patients treated between 3 and 5 pm, as shown in Figure 5?
4. Please show references in p12 (line 342-3 and line 348-9).
5. To exclude the possibility of off-target effects, the authors should perform the siRNA experiments in vitro, using at least two siRNA target sequences.
Author Response
Reply:
Thank you for your thoughtful feedback. In response to your comments, I have made modifications to our manuscript and conducted additional experiments. I have addressed each of your comments and summarized the corresponding changes in the following sections for comments 1 through 5.
Comment 1. Why did you select UM-UC-3 and J82 cells? These cell lines were derived from patients with muscle-invasive bladder cancer (MIBC). PDD-TURBT is especially useful for detecting non-muscle invasive bladder cancer (NMIBC).
Reply to Comment 1:
We selected the UM-UC-3 and J82 cell lines for this experiment despite their origin from muscle-invasive bladder cancer. These cell lines were chosen due to their high tumorigenicity and ease of engraftment into nude mice. Ideally, a naturally induced bladder cancer model using agents such as BBN would be more representative, but we consider this an important area for future research. In response to your comment, we have included this consideration in the manuscript (page 26, lines 395–397).
Comment 2. Figure 4 suggests that the intracellular PPIX accumulation is associated with CPOX and FECH levels through circadian rhythm. Did the authors evaluate the association between PER2/BMAL1 and CPOX/FECH levels in vitro?
Reply to Comment 2:
While we evaluated the association between intratumoral PpIX concentration and heme synthesis pathway enzymes using xenograft mouse models, we did not conduct similar evaluations in vitro. However, we observed a correlation between CPOX, FECH, and clock genes in vivo, which we believe provides more robust reproducibility than in vitro experiments. In mammals, where the on-and-off states of clock genes are regulated by light stimulation, we were able to demonstrate these phenomena.
Comment 3. The previous studies have shown that the circadian rhythm is impaired in cancer. Why were the fluorescence intensities highest in patients treated between 3 and 5 pm, as shown in Figure 5?
Reply to Comment 3:
You are correct that some studies have reported impaired circadian rhythms in cancer. However, other reports, such as Jia et al. (Oncol Rep. 2020; 45:1033–1043), have demonstrated the presence of circadian rhythms in clock genes within UM-UC-3 cells. In our study, we observed circadian shifts in the expression of clock genes PER2 and BMAL1. Furthermore, we identified a potential link between the circadian rhythms of PpIX accumulation and the heme synthesis pathway. As shown in Figure 5, the highest fluorescence intensity observed between 3 and 5 pm may reflect these rhythms. However, this finding was limited to specific cell lines and may not be generalizable to all bladder cancers. To address this limitation, future clinical studies are needed to evaluate the timing-related differences in intratumoral PpIX concentration during PDD-TURBT. This significant limitation has been noted in the revised manuscript (page 26, lines 397–400).
Comment 4. Please show references in p12 (line 342-3 and line 348-9).
Reply to Comment 4:
We sincerely apologize for the missing references in the sections you mentioned. We have added the appropriate citations to the manuscript as indicated.
Comment 5. To exclude the possibility of off-target effects, the authors should perform the siRNA experiments in vitro, using at least two siRNA target sequences.
Reply to Comment 5:
Thank you for this important comment. As you noted, off-target effects are a concern with siRNA experiments. To address this, we conducted additional experiments using multiple siRNAs targeting PER2 and BMAL1 to minimize off-target effects. Specifically, we prepared two additional siRNAs for each gene, resulting in a total of three siRNAs for both PER2 and BMAL1. We measured intracellular PpIX concentration and relative mRNA expression levels over 24 hours in both UM-UC-3 and J82 cells. These results, summarized in the revised version of Figure 2, show that no circadian rhythms were observed with any of the siRNAs, and relative mRNA levels of PER2 and BMAL1 were consistently knocked down throughout the day in both cell lines. This minimizes the possibility of off-target effects and strengthens the reproducibility of our findings on the association between clock genes and PpIX concentration. We have updated the manuscript accordingly (page 14, lines 249–252; page 16, lines 259–272; page 24, lines 353–355; and revised Figure 2).
Finally, we sincerely appreciate your insightful feedback, which has significantly improved the quality of our manuscript.
Round 2
Reviewer 3 Report
Comments and Suggestions for Authors
There is no additional comment.